# Yorkshire Lung Screening Trial (YLST): protocol for a randomised controlled trial to evaluate invitation to community-based low-dose CT screening for lung cancer versus usual care in a targeted population at risk

Philip AJ Crosbie [1], Rhian Gabe,[2] Irene Simmonds,[3] Martyn Kennedy,[4] Suzanne Rogerson,[5] Nazia Ahmed,[3] David R Baldwin [6], Richard Booton,[7] Ann Cochrane,[8] Michael Darby,[9] Kevin Franks,[10] Sebastian Hinde [11], Sam M Janes,[12] Una Macleod,[13] Mike Messenger,[14] Henrik Moller,[15] Rachael L Murray [16], Richard D Neal,[3] Samantha L Quaife,[17] Mark Sculpher,[11] Puvanendran Tharmanathan,[8] David Torgerson,[8] Matthew EJ Callister[4]

For numbered affiliations see end of article.

**Correspondence to**
Dr Matthew EJ Callister;
matthew.callister@nhs.net

## ABSTRACT

**Introduction** Lung cancer is the world's leading cause of cancer death. Low-dose computed tomography (LDCT) screening reduced lung cancer mortality by 20% in the US National Lung Screening Trial. Here, we present the Yorkshire Lung Screening Trial (YLST), which will address key questions of relevance for screening implementation.

**Methods and analysis** Using a single-consent Zelen's design, ever-smokers aged 55–80 years registered with a general practice in Leeds will be randomised (1:1) to invitation to a telephone-based risk-assessment for a Lung Health Check or to usual care. The anticipated number randomised by household is 62 980 individuals. Responders at high risk will be invited for LDCT scanning for lung cancer on a mobile van in the community. There will be two rounds of screening at an interval of 2 years. Primary objectives are (1) measure participation rates, (2) compare the performance of PLCO$_{M2012}$ (threshold ≥1.51%), Liverpool Lung Project (V.2) (threshold ≥5%) and US Preventive Services Task Force eligibility criteria for screening population selection and (3) assess lung cancer outcomes in the intervention and usual care arms. Secondary evaluations include health economics, quality of life, smoking rates according to intervention arm, screening programme performance with ancillary biomarker and smoking cessation studies.

**Ethics and dissemination** The study has been approved by the Greater Manchester West research ethics committee (18-NW-0012) and the Health Research Authority following review by the Confidentiality Advisory Group. The results will be disseminated through publication in peer-reviewed scientific journals, presentation at conferences and on the YLST website.

**Trial registration numbers** ISRCTN42704678 and NCT03750110.

### Strengths and limitations of this study

► Prospective evaluation of three criteria for screening population selection to inform the optimal approach for screening implementation.

► The Zelen's study design and unscreened control group (current standard of care in the UK) will allow us to assess the impact of lung cancer screening at a population level.

► Prospective evaluation of participation rates and clinical outcomes of community-based lung cancer screening with biennial low-dose CT scans.

► A nested substudy within Yorkshire Lung Screening Trial (the Yorkshire Enhanced Stop Smoking Trial) will evaluate the effectiveness of a fully integrated colocated stop smoking service in a screening programme.

► The study is not powered to assess the impact of screening on lung cancer specific mortality.

## INTRODUCTION

Lung cancer is the leading cause of cancer mortality nationwide, responsible for approximately 30 000 deaths/year in England and Wales. Low-dose computed tomography (LDCT) screening reduced mortality in the US National Lung Screening Trial (NLST), which randomised 53 439 participants at high risk to annual LDCT or chest X-ray screening for 2 years.[1] LDCT screening reduced lung cancer specific and all-cause mortality by 20% and 6.7%, respectively. This finding has recently been confirmed by the Nederlands-Leuvens Longkanker Screenings Onderzoek

(NELSON) trial investigators.[2] The Multicentric Italian Lung Detection (MILD) trial demonstrated a 39% reduction in lung cancer specific mortality over 10 years with LDCT screening.[3] In 2014, the US Preventive Services Task Force (USPSTF) recommended LDCT screening for lung cancer based on the NLST findings, extending the upper age to 80 years.[4 5] Despite on-going implementation in the USA, several important issues remain unresolved, including optimising identification of high-risk individuals for screening, embedding smoking cessation in screening programmes and improving uptake in those at highest risk.[6]

## Screening selection criteria

Within NLST, very few deaths were prevented in the two lowest lung cancer risk quintiles.[7] In addition, only 27% of patients diagnosed with lung cancer in the USA[8] and 35% in Yorkshire would have been eligible for screening by NLST criteria.[9] Composite risk prediction tools show improved sensitivity, specificity and positive predictive value when retrospectively compared with NLST criteria.[10] Such tools may, therefore, identify a greater number of people with lung cancer, and so improve the effectiveness of screening. However, they also tend to identify an older cohort, and thus some of the effect on life years gained, and therefore cost-effectiveness may be attenuated.[11]

The Yorkshire Lung Screening Trial (YLST) will prospectively assess three proposed methods of screening population selection: USPSTF, Liverpool Lung Project (LLP) and Prostate, Lung, Colorectal and Ovarian (PLCO) Cancer Screening Trial models. The elgibility criteria for screening in the NLST were age 55–74, ≥30 pack years and smoked within 15 years.[1] The USPSTF concluded that screening should be extended to age 80[12] and issued a recommendation in 2014.[4 5] The $PLCO_{M2012}$ risk prediction model was derived in 80375 participants of the PLCO Study, and validated in a separate cohort of 37332.[13] Lung cancer risk ≥1.51% over 6 years was identified as the threshold defining consistent reduced mortality with LDCT screening.[10] The UK Lung Cancer Pilot Screening Trial (UKLS) used the LLP model set at a threshold of ≥5% over 5 years to select participants for screening.[14] YLST will prospectively assess the performance of USPSTF criteria, the $PLCO_{M2012}$ risk prediction model (6-year risk threshold ≥1.51%) and LLP (V.2) model (5-year risk threshold ≥5%).

## Optimising participation for those at most risk

Lung cancer risk is the highest within lower socioeconomic status (SES) communities.[15] Both low SES and current smoking are associated with lower participation in cancer screening programmes[16 17] and research studies.[18–20] In UKLS, transport difficulties were the most commonly reported practical barrier to participation[21] especially in lower socioeconomic quintiles, while emotional barriers including higher affective risk perceptions were more frequently reported by smokers. Attendees of Manchester's Lung Health Check (LHC)

pilot expressed a preference for community-based screening.[22] A randomised trial of mobile mammography for breast screening showed increased participation rates especially in older and low-income populations.[23] Interventions shown to improve participation in colorectal screening include a primary care endorsement letter, advanced notification of the screening offer and reminder reinvitations.[24–28] Enhanced participant information leaflets as well as invitations targeted to address psychological barriers are also promising,[29] and in the Lung Screen Uptake Trial a low information burden, targeted and stepped invitation approach improved uptake in the lowest SES quintile.[30] These factors form part of our invitational strategy.

## Cost-effectiveness of LDCT screening

LDCT screening needs to ensure the considerable financial investment results in the maximum healthcare gain. Previous UK studies have estimated the incremental cost-effectiveness ratio (ICER) of screening as £8466 per quality-adjusted life year (QALY) in the UKLS,[14] £10 069/QALY for mobile LDCT screening in Manchester[31] and £28 169/QALY in a systematic review-based analysis.[32] The resultant variation in conclusions regarding the cost-effectiveness of screening, driven by different model approaches and populations, highlights the need for further research. By assessing screening yield using different risk tools and thresholds, it will be possible to compare both the clinical and cost-effectiveness of different selection criteria.

## RATIONALE

Questions remain about whether the mortality reduction demonstrated in NLST can be translated into routine clinical care. Implementation studies are required to demonstrate participation among people at highest risk of lung cancer particularly from deprived populations where the burden of disease is greatest. Basing screening in convenient community locations is one proposed strategy to increase uptake, especially in deprived populations.[22] Invitation to an 'LHC' not 'lung cancer screening' is an approach that has been successfully used to increase participation in the previous UK studies leading to screening of higher risk individuals.[33–35] Furthermore, clarification of the optimal strategy for defining a high-risk population for screening in the UK would aid any subsequent roll-out of a national screening programme.

## OBJECTIVES

The primary objectives of the study are:
1. To measure participation rates of a community-based lung cancer screening programme.
2. To compare the performance of USPSTF, $PLCO_{M2012}$ (≥1.51%) and LLP (V.2) (≥5%) criteria for identifying individuals at high risk of lung cancer for LDCT screening.

3. To assess the clinical outcomes of invitation to targeted community-based LDCT screening for lung cancer versus usual care (no invitation).

The study's secondary objectives are:

1. To undertake a health economic evaluation of community-based LDCT screening for lung cancer, specifically comparing the health economic impact of the three selection criteria (USPSTF, PLCO and LLP).
2. To evaluate the performance of the screening programme.
3. To determine the effect of invitation to a screening programme on smoking rates comparing intervention and usual care arms.

## OUTCOME MEASURES

The primary outcome measures are:

► The proportion of the study population allocated to intervention who undergo telephone assessment and are screened according to the USPSTF, PLCO and LLP criteria.
► Lung cancers identified in participants selected for LDCT screening according to the USPSTF, PLCO and LLP criteria over two rounds of biennial screening.
► The incidence of advanced lung cancer in the intervention and control arms over the course of the study.

The secondary outcome measures are:

► ICER for community-based lung cancer screening overall and according to the three criteria for identifying candidates for screening.
► Screening performance including the following parameters:
  – Cancer detection rate and number needed to screen (NNS) to detect one lung cancer according to the risk criteria over two screening rounds.
  – False-positive and false-negative rate.
  – Rate of investigation of benign disease and benign surgical resection rate.
  – Attendance by LDCT screening round and according to participant characteristics (age, sex, smoking status, ethnicity and SES).
  – Treatment of screen detected lung cancer including surgical resection rate.
  – Investigations, treatments and adverse events generated from screening including incidental findings.
  – Interval cancers and recall rates in those undergoing screening.
► Smoking prevalence at start and end of the study in intervention arm and usual care arm.
► Participation rates in telephone assessment by age, sex, smoking status, ethnicity and SES.
► Route to diagnosis, histological subtype, stage and treatment of lung cancers in the intervention (LDCT screened group, eligible respondents, non-respondents and ineligible low risk responders) and usual care arms.
► Lung cancer and all-cause mortality by trial arm.

► Numbers of nodules detected, proportion with eventual diagnosis of cancer by: size (volume), PanCan Malignancy Risk Prediction Score[36] and volumetry-derived volume doubling time.
► Prevalence of undiagnosed airflow obstruction and coronary artery calcification in the screened population.
► Quality-of-life (EQ-5D and SF-12) scores.[37 38]

## METHODS

### Study design

A two-arm, implementation study using a single-consent Zelen's randomised controlled design.[39] Ever smokers, age 55–80, are randomised to either invitation to a telephone assessment followed by a community-based LHC if at high risk (according to any of USPSTF, PLCO and LLP criteria) or usual care (no invitation) (figure 1).

### Setting

YLST is conducted within the catchment area of a single secondary care site—Leeds Teaching Hospitals (LTH). LHCs take place in mobile units at convenient community locations. A single mobile unit (comprising a mobile CT scanner and support accommodation) rotates through community locations across Leeds on a monthly basis.

### Participant identification

Potential participants are identified from participating general practices (GPs) based within Leeds Clinical Commissioning Group (CCG). It is anticipated that ≈80 practices will take part. Eligible participants with records indicating they have ever smoked will be identified from groups of GPs on a monthly basis during the 2-year recruitment period (November 2018 to October 2020).

### Randomisation

To avoid cohabitees being allocated to different arms of the study, the unit of randomisation is the household. A 1:1 randomisation programme using simple randomisation without stratification is run monthly to allocate approximately half the households to either the intervention or control arm.

### Randomisation cohort eligibility criteria

#### Inclusion

► Registered with a participating GP.
► Age 55–80 years (inclusive) at time of data extraction.
► Registered as current or ex-smoker in primary care databases.

#### Exclusion

► Malignant neoplasm of trachea, bronchus, lung, thymus or pleura diagnosed within 5 years.
► Any previous diagnosis of metastatic cancer.
► On palliative care register (Gold Standards Framework).
► Primary care coded diagnosis of dementia.
► Registered types 1 and 2 objection to participation in the GP Extraction Service.

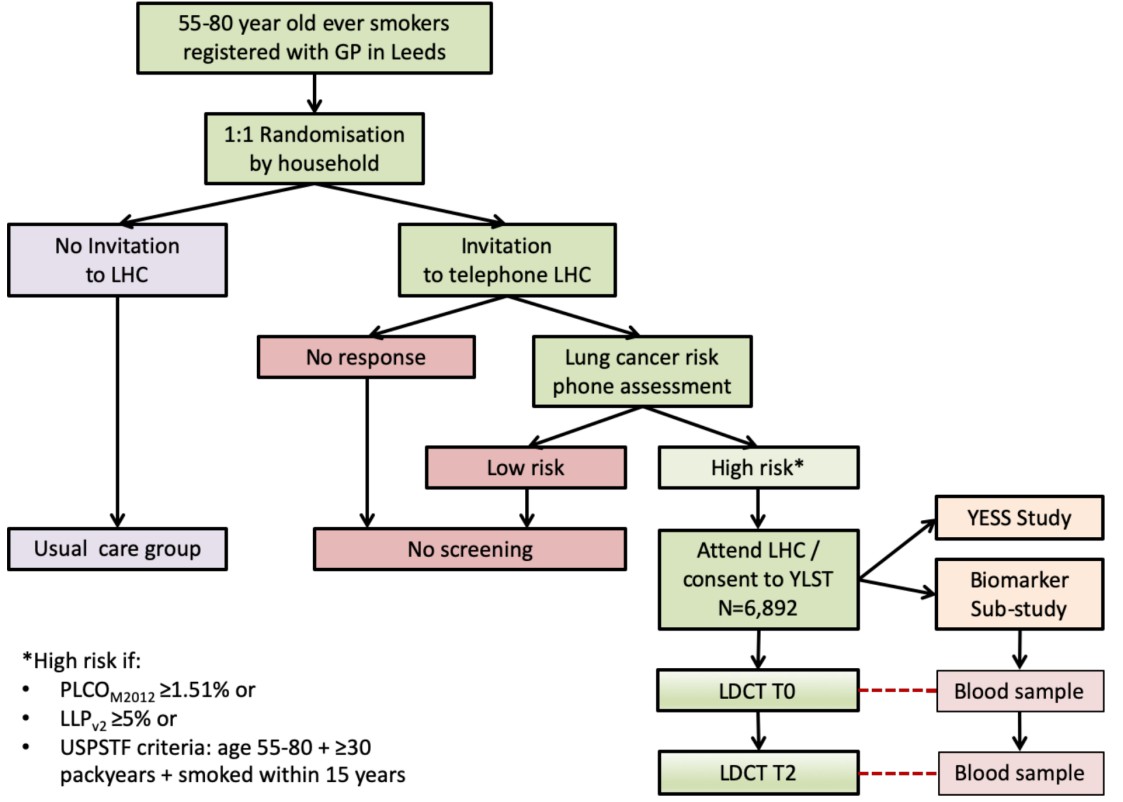

**Figure 1** YLST flow diagram. GP, general practice; LDCT, low-dose computed tomography; LHC, Lung Health Check, LLP, Liverpool Lung Project; PLCO, Prostate, Lung, Colorectal and Ovarian; USPSTF, US Preventive Services Task Force; YESS, Yorkshire Enhanced Stop Smoking; YLST, Yorkshire Lung Screening Trial.

▶ GP coded diagnosis of 'severe frailty' or recorded Electronic Frailty Index >0.36.
▶ Nursing home residents or GP coded status of 'Housebound'.
▶ CT thorax within 12 months of data extraction (including high resolution CT thorax, CT thorax (±abdomen/pelvis) with contrast, CT pulmonary angiogram).

### Invitation procedure and adherence

Participants in the intervention arm are invited to contact a telephone LHC assessment to check eligibility and to book an appointment. The approach schedule is staggered over a month and includes: a preinvitation letter, notifying individuals of the LHC service, followed by an invitation letter and up to two reminders for non-responders. All letters have the GP digital signature/letterhead. The invitation is accompanied by a low-burden information leaflet designed to address psychological barriers to participation. This stepped, low-burden and targeted approach was favourably reviewed by the Lung and Mesothelioma Patient Support Group at LTH Trust; it uses material developed in response to research in low SES populations[40] and leaflets used in the Lung Screen Uptake Trial.[35]

### Telephone triage

Invitees who contact the telephone triage are assessed for screening eligibility according to the USPSTF criteria,

and future lung cancer risk using $PLCO_{M2012}$ and LLP (V.2) models. Individuals fulfilling at least one of these three criteria are invited to attend an LHC. Participants not eligible for screening receive a 'Keeping your lungs healthy' leaflet with general advice regarding respiratory health. Current smokers are offered referral to the Leeds Stop Smoking Service.

### Lung Health Check

The LHC is a nurse-led service. On arrival, attendees watch a short film describing the LHC process, explaining the context of the study and discussing the benefits and harms of LDCT screening for lung cancer (eg, overdiagnosis, invasive tests for benign disease, worry about screening results and radiation exposure). A full participant information sheet is also provided. Fully informed written consent for study participation is obtained. LHC components include:
▶ LHC questionnaire.
▶ Measurement of height and weight, spirometry, oxygen saturation and exhaled carbon monoxide.
▶ Smoking cessation intervention.
▶ Biomarker substudy.
▶ LDCT scan—undertaken immediately or at a future date, determined by participant preference.

The CT scan is requested by a doctor or research nurse with appropriate Ionising Radiation (Medical Exposure) Regulations accreditation. LHC and screening results

are communicated to participants and primary care within 4weeks. Participants with obstructive spirometry without a prior diagnosis of Chronic Obstructive Pulmonary Disease or asthma are referred to the Community Respiratory Team.

### Smoking cessation provision

Unless explicitly declined, current smokers (smoked within 4 weeks and/or carbon monoxide reading >6 parts/million) see a specialist smoking cessation practitioner. Support is provided in line with National Institute for Health and Care Excellence Public Health guideline 48 and comprises one session of behavioural support and provision of pharmacotherapy at the time of the LHC. Pharmacotherapy may include Nicotine Replacement Therapy provided through delegated prescribing, a commercially available e-cigarette, or a varenicline or bupropion prescription to be taken to the participant's GP. Follow-up is face to face or by telephone, typically weekly for up to 12 weeks. The effectiveness and acceptability of the service plus that of a personalised intervention is being evaluated in the Yorkshire Enhanced

Stop Smoking Study, a nested substudy within YLST (ISRCTN63825779).[41]

### Biomarker substudy

YLST participants are invited to take part in the biomarker substudy. This is optional and involves blood samples taken at each visit to the support vehicle and in the cancer clinic for those with a positive scan. Samples are transferred to collaborating laboratories (commercial and academic) for analysis. The overarching aims of the biomarker study are to investigate whether biomarkers can improve the efficacy of risk stratified population selection, pulmonary nodule management and screening intensity.

### LDCT screening

YLST participants are offered biennial LDCT screening over two rounds. The baseline round is termed T0 and the second round T2. Pulmonary nodule surveillance may occur 3 months and/or 12 months after baseline— termed T0 +3/12 and T0 +12/12, respectively. In the second screening rounds, these timepoints are T2 +3/12 and T2 +12/12 (figure 2).

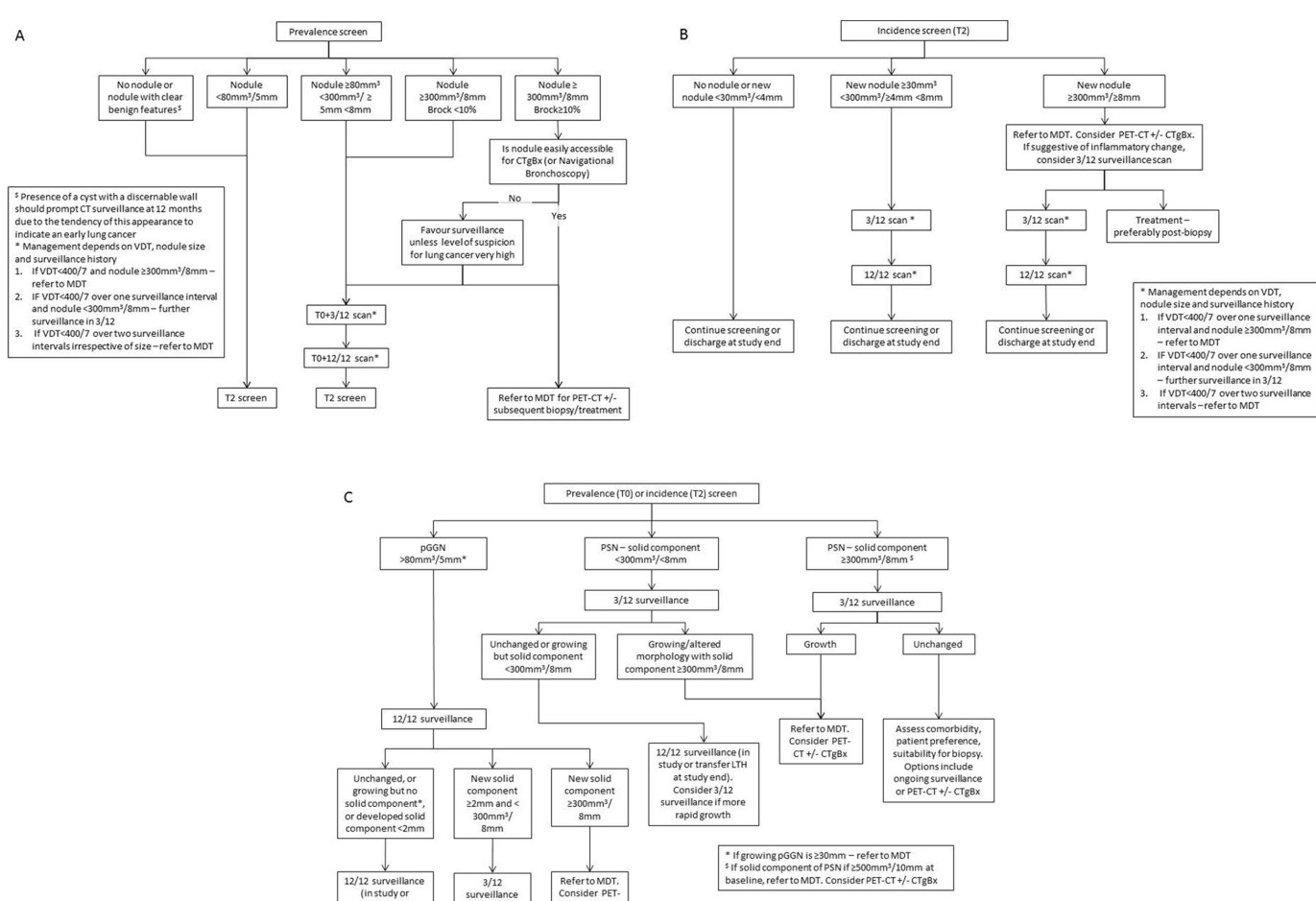

**Figure 2** Management algorithms for (A) solid pulmonary nodules detected in the first screening round (T0); (B) solid nodules detected in an incidence round (T2) and (C) subsolid nodules detected during screening. CTgBx, CT-guided percutaneous biopsy; LTH, Leeds Teaching Hospitals; MDT, lung cancer multidisciplinary team meeting; PET, positron emission tomography; pGGN, pure ground glass nodule; PSN, part solid nodule; VDT, volume doubling time.

## LDCT scan

A sixteen-channel (or higher) mobile multidetector CT will be used throughout the study. Participants lie supine with arms above their head and thorax in the midline. Imaging is performed during suspended maximal inspiration. No intravenous contrast is administered. The lung parenchyma (lung apices to bases) is scanned in its entirety in a single craniocaudal acquisition. Radiation exposure is minimised while maintaining good image quality (effective radiation dose <2 mSv). Image reconstruction is standardised and used for any subsequent follow-up examinations. Images are transferred securely from the mobile unit to LTH picture archiving and communication system.

## CT scan reporting

CT scans are reported by a consortium of consultant radiologists with a specific interest in thoracic imaging. All radiologists have substantive appointments at National Health Service hospitals and contribute regularly to Lung Cancer MultiDisciplinary Team meetings. Scans are reported using Veolity (MeVis Medical Solutions AG), a bespoke software package for lung cancer screening including automated volumetry and computer-aided detection. The same software will be used throughout the study. Reports are categorised as either negative, indeterminate or positive; each scan may also have an additional 'incidental' finding label. Categories are defined as follows:

▶ Negative: normal scan or abnormal scan that does not require further investigation or intervention.
▶ Indeterminate: indeterminate pulmonary nodule(s) needing surveillance.
▶ Positive: finding(s) concerning for lung cancer requires immediate investigation in the Fast Track Lung Cancer Clinic.
▶ Incidental: other finding(s) that requires clinical review.

Individuals with a negative LDCT scan at T2 are discharged from the study. A false positive is defined as an individual who has been investigated for possible lung cancer in the Fast Track Lung Cancer Clinic but is eventually found not to have lung cancer.

## Quality assurance

A random selection of negative scans (≈5%) is second read. Any discordance between scan reports is resolved in the screening review meeting.

## Screening review meeting

Screening review meeting comprises a consultant radiologist, consultant respiratory physician and a study nurse or senior clinical trials assistant. Indeterminate, positive and incidental scans are routinely discussed; second read negative CT scans with discordant reports and others flagged for clinical reasons are also discussed.

## Pulmonary nodule management algorithms

Nodule management is based on the British Thoracic Society guidelines,[42] modified for a screening programme.

Nodules are categorised solid or subsolid nodule (SSN). SSN are either pure ground glass nodule (pGGN) or part solid nodule. Nodules are measured using volumetry. The risk of malignancy is determined using the Brock risk calculator.[36] Surveillance for indeterminate pulmonary nodules occurs at 3 and 12 months after detection in most cases, or 12 months from baseline for pGGNs or cysts with discernible walls. Occasionally the surveillance schedule is modified; this will be determined on a case-by-case basis through the screening review meeting.

## STATISTICAL CONSIDERATIONS

### Power

The primary outcome measure for assessing participation is the proportion of invitees responding to telephone invitation. The anticipated number randomised by household is 62 980 individuals and with 31 490 in the intervention arm this will be reported with high precision. With 14 170 invitees anticipated to undergo assessment by the risk criteria, there is >95% power to detect differences in the proportion being 'risk positive' using pairwise, two-sided McNemar tests at the 5% significance level. Expected values are 24%, 41% and 29% for USPSTF, PLCO and LLP, respectively, and a correlation of at least 0.3 (based on a survey of ever smokers aged 55–80 conducted in Leeds CCG). Responders are invited to the LHC with LDCT screen if they are positive on at least one of the risk criteria and on this basis, 6892 are expected to take part in the screening programme.

The primary outcome measure for USPSTF, PLCO and LLP criteria performance is number of LDCT screen-detected lung cancers; defined as cancers diagnosed following an invited LDCT screen (prevalence or incidence) or follow-up of an abnormality detected at these screens. UKLS used the LLP (V.2) risk tool with predicted 5-year risk ≥5% as an inclusion criteria for randomisation to invitation to screen and 2.1% were diagnosed with lung cancer within 12 months.[14] Data from NELSON suggest that a similar proportion of cancers are detected at the incidence screening round.[43] YLST aims to maximise recruitment of those at high risk (n≈6892) and we expect to observe 289 screen-detected lung cancers with a detection rate of 4.2% after 4 years. This does not include interval cancers for which there is little prior data but few such cancers are anticipated.

With 289 screen-detected lung cancers, there is >95% power to detect differences in the number of cancers detected using pairwise, two-sided McNemar tests for the comparison of USPSTF versus PLCO and LLP versus PLCO at the 5% significance level, with expected values of 212 cancers detected by USPSTF (73.5%), 276 cancers detected by PLCO (95.5%) and 225 cancers detected by LLP (78%). The expected correlation among screen-detected cancers is low (0.05 with USPSTF vs LLP and 0.1 for other comparisons). These figures are based on a survey of 71 cancers in ever smokers aged 55–80 conducted in LTH.[9] Since there is a much smaller difference in the

number of cancers detected between the USPSTF and LLP criteria and low correlation, anticipated power to reliably detect a difference is low (25%).

### Power to detect differences in advanced lung cancer

The YLST design allows us to evaluate the effect on lung cancer outcomes of (i) invitation to risk assessment followed by screening in ever smokers aged 55–80 (the randomisation cohort) and (ii) LDCT screening in those eligible at risk assessment. The primary outcome for the comparison of targeted community-based LDCT screening for lung cancer versus usual care in YLST is incidence of advanced lung cancer in the randomisation cohort. Incidence of lung cancer and proportion of advanced disease (stage III or IV) will be reported in the intervention and control groups as a whole, in the expected 17 320 non-respondents, 7278 ineligible low risk and 6892 eligible respondents within the intervention group. Breakdown of late-stage cancer by non-risk, low-risk and high-risk respondents will be used to adjust for observations in the control group and allow evaluation of (ii) using adapted methods for control of non-compliance.[44] We will estimate the effect of the distribution on stage after the first 4 years.

Assuming a lung cancer incidence of 5, 2 and 3 per thousand per year in the eligible, non-eligible and non-respondents, the overall cumulative incidence over 4 years would be 12.82 per thousand. Due to randomisation, we would expect the same in the controls (with slightly higher incidence in the intervention group due to lead time and overdiagnosis). From national rates we would expect 75% of symptomatic cancers to be at stage III or IV compared with 14% in UKLS. Assuming that 25% of all tumours in the screened population (screen detected and symptomatic) are stage III or IV, and this is 75% in non-responders or those at low risk, then the incidence of cancers at stage III or worse in the intervention group would be 7.43 per thousand. Suppose we observe a cumulative 4-year incidence of stage III or worse of 9.62 per thousand in the 31 490 in the control group (75% of 12.82), then the power to detect this difference (two-sided testing 5% significance level) is 85%. The relative risk (RR) estimate for late-stage cancers would be 0.77 (95% CI: 0.60 to 0.94), a 23% reduction.

In the LDCT screened group (anticipated n=6892), we estimate that there will be a cumulative 4-year rate of advanced stage cancers of $0.005 \times 4 \times 0.25 = 0.005$. In the control group, due to randomisation, we infer an identical population of non-eligible and eligible potential responders. Assuming that 65% of all cancers in populations unexposed to screening are advanced including in the control group, then in the group of 6892 potential eligible responders in the control arm, we anticipate a cumulative 4-year incidence of advanced cancers of $0.005 \times 4 \times 0.65 = 0.013$. If the latent eligible potential responder group were directly observed, 6892 subjects per group would confer 99% power to detect the difference in rates of advanced stage disease as significant (5%

significance level, two-sided testing). However, as it is imputed by subtraction of the quantities observed for the unscreened populations in the intervention arm, the SE of the comparison will be inflated by around 15%, and therefore we will have nearer 85% power. For calculation of estimated outcomes in the latent group and its SE, see Cuzick *et al*.[44]

### Analysis

Baseline characteristics by trial arm and a diagram depicting the number and flow of patients through the trial will be presented. Statistical tests will be two sided using a 5% significance level unless otherwise specified and 95% CIs will be reported as appropriate.

#### Primary analyses

Descriptive analyses will be used to present the proportion of the population undertaking telephone risk assessment who are screened according to the USPSTF, PLCO and LLP criteria over two rounds of biennial screening. The number of screen-detected lung cancers over 4 years will be compared by risk criteria. Pairwise, two-sided McNemar tests will be used to compare differences between the risk criteria in terms of proportions of cancers detected that were above the risk threshold. Stage distribution and numbers of advanced lung cancers (stage III or IV) will be reported in the intervention and control groups as a whole after the planned two rounds of screening. Cumulative incidence of advanced lung cancer over this period in those allocated to invitation to targeted community-based LDCT screening versus those allocated to no invitation will be compared through estimation of the RR and associated 95% CIs using Poisson regression.

#### Secondary analyses

Simple descriptive analyses will be used to present data for secondary outcomes including participation rates, characteristics of those attending LDCT screening and adverse events. The NNS to detect one lung cancer for each of the three risk criteria over the two rounds of screening will be reported. To calculate NNS, total number of eligible, risk-positive participants attending an LDCT scan on the mobile van will be divided by total number of screen-detected cancers. NNS to detect one lung cancer is the reciprocal of the cancer detection rate. The performance of the risk prediction tools will also be investigated in terms of 4-year cumulative incidence of lung cancer for those found eligible and not eligible for LDCT screening. In addition, the performance of the risk models will be assessed for screen-detected early stage cancers.

#### Health economic analysis

A health economic analysis will be conducted with the primary aim of determining the cost-effectiveness of the YLST screening programme. Secondary analysis will explore the expected cost-effectiveness if such an approach was expanded to a national setting, reflecting the worse current lung cancer rates and outcomes in the

Yorkshire region, and determine the relative costs and benefits associated with the different screening criteria. Analysis of the uncertainty around the cost-effectiveness estimates will be conducted including scenario and probabilistic sensitivity analysis.

## ETHICS AND DISSEMINATION

YLST is an implementation study, designed to replicate a possible approach strategy were lung cancer screening to be introduced in the UK. This involves framing the intervention as an 'LHC' service without specific mention of research in the original approach material. However, the patient population at participating GP surgeries are informed of the research through dissent and opt out posters displayed at the surgery prior to data extraction. This is the only contact with a non-invited usual care control population. Those attending an LHC receive a full explanation from the research nurse/senior clinical trial assistant about screening (including the harms and benefits), and the context of LHCs as a health intervention being studied as part of research. Data collection and LDCT screening only proceed following informed consent. In addition, we wish to compare outcomes to the non-invited control group to allow assessment of lung cancer stage and smoking rates between the two arms. With longer follow-up, we wish to determine overdiagnosis rates and the effect on lung cancer and all-cause mortality. Participants found to be at lower risk, who do not meet with a research nurse, are sent information about how their data will be used and details of how to opt out. We will track outcomes in three groups who have not consented to study participation: a non-invited usual care control population, invited non-responders and those contacting the telephone triage service but not fulfilling the criteria for LDCT screen. This study design has been approved by the Health Research Authority following review by the Confidentiality Advisory Group in order to collect and process this confidential data by a Section 251 exemption. This approach was also reviewed and supported by a patient group recruited through the Yorkshire Cancer Patient Forum.

Trial governance will be supported by a Trial Management Group (TMG), an Independent Data Monitoring Committee (IDMC) and an Independent Trials Steering Committee (TSC). The TMG will oversee the day-to-day running and progress of the trial. The IDMC is an advisory body to confidentially review interim data and safety and can recommend premature closure or modification of the trial to the TSC. The TSC provides oversight and will consider reports from the TMG and IDMC, as well as external sources. The TSC make the final decision to recommend prematurely closing the trial if considered necessary. The independent members of these committees include experts in the field of cancer screening, respiratory medicine, radiology and statistics, and a patient/public representative sits on the TSC. Findings from the study will be written in accordance with the CONSORT

Statement,[45] submitted for publication to relevant peer-reviewed journals and presented at conferences. A summary of results will be provided for participants on the study website. An independently chaired Biomarker Committee provides oversight to the biomarker substudy including the assessment of expressions of interest from potential collaborators.

## DISCUSSION

YLST will address a number of important issues directly relevant to the implementation of lung cancer screening. The UK does not currently have a national lung cancer screening programme, and thus the control group in YLST receive usual care. Despite convincing evidence of mortality reduction in large randomised controlled trials, previous studies have not demonstrated improved lung cancer outcomes across a whole population invited for screening. YLST will assess lung cancer outcomes in the whole intervention population (people undergoing LDCT screening, people at lower risk not eligible for screening and non-responders) compared with the non-invited control population. YLST will also assess what effect, if any, invitation to LHC has on smoking rates across the intervention and control populations. Previous studies have shown conflicting effects of screening on smoking behaviours[46–48] and typically offered low levels of cessation support. However, such comparisons are subject to bias related to the characteristics of people who are likely to participate in research studies where full consent was required for all participants.

**Author affiliations**
[1]Division of Infection, Immunity and Respiratory Medicine, The University of Manchester, Manchester, UK
[2]Centre for Cancer Prevention, Wolfson Institute of Preventive Medicine, Queen Mary University of London, London, UK
[3]Leeds Institute of Health Sciences, University of Leeds, Leeds, UK
[4]Department of Respiratory Medicine, Leeds Teaching Hospitals NHS Trust, Leeds, UK
[5]Department of Research and Innovation, Leeds Teaching Hospitals NHS Trust, Leeds, UK
[6]Department of Respiratory Medicine, City Campus, Nottingham University Hospitals, Nottingham, UK
[7]Lung Cancer and Thoracic Surgery Directorate, Heart and Lung Division, Manchester University NHS Foundation Trust, Manchester, UK
[8]York Trials Unit, Department of Health Sciences, University of York, York, UK
[9]Department of Radiology, Leeds Teaching Hospitals NHS Trust, Leeds, UK
[10]Leeds Cancer Centre, Leeds Teaching Hospitals NHS Trust, Leeds, UK
[11]Centre for Health Economics, University of York, York, UK
[12]Department of Respiratory Medicine, University College London, London, UK
[13]Hull York Medical School, University of Hull, Hull, UK
[14]Leeds Centre for Personalised Medicine and Health, University of Leeds, Leeds, UK
[15]Thames Cancer Registry, Kings College London, London, UK
[16]Division of Epidemiology and Public Health, Faculty of Medicine, University of Nottingham, Nottingham, UK
[17]Research Department of Epidemiology and Public Health, University College London, London, UK

**Contributors** All authors contributed to the development and set up of the study. MC and PAJC were responsible for the overall conduct of the study. MC, SR, MD,

MK and IS were responsible for the day-to-day running of the study. SR led the Lung Health Check (LHC) team. MD provided oversight for study thoracic radiology. MC, PAJC, RG, DB, HM, PT, SLQ, SH, IS and RDN developed the protocol. RDN and UM provided guidance around the primary care aspects of the study. RLM designed the smoking cessation provision in the study. RB and PAJC contributed the design of the LHC. SLQ and SJ contributed to the patient facing approach literature used. RG and DT developed the study's statistical/analysis plan. SH and MS developed the health economic study design. IS and AC provided project management and NA data management. PAJC, MM and KF were responsible for the conduct of the biomarker study. PAJC, RG and MC wrote the manuscript, all authors contributed to, reviewed and approved the final version of the manuscript.

**Funding** This work was funded by the Yorkshire Cancer Research (Award references L403 & L403B).

**Competing interests** PAJC has received consultation fees and shares options from Everest Detection. PAJC is supported by the NIHR Manchester Biomedical Research Centre.

**Patient and public involvement** Patients and/or the public were involved in the design, or conduct, or reporting, or dissemination plans of this research. Refer to the Methods section for further details.

**Patient consent for publication** Not required.

**Provenance and peer review** Not commissioned; externally peer reviewed.

**ORCID iDs**
Philip AJ Crosbie http://orcid.org/0000-0001-8941-4813
David R Baldwin http://orcid.org/0000-0001-8410-7160
Sebastian Hinde http://orcid.org/0000-0002-7117-4142
Rachael L Murray http://orcid.org/0000-0001-5477-2557

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
