## [Reviewer comments · BMJ Open]

ARTICLE DETAILS

TITLE (PROVISIONAL)	The Yorkshire Lung Screening Trial (YLST): protocol for a randomised controlled trial to evaluate invitation to community-based low dose computed tomography screening for lung cancer versus usual care in a targeted population at risk.
AUTHORS	Crosbie, Philip; Gabe, Rhian; Simmonds, Irene; Kennedy, Martyn; Rogerson, Suzanne; Ahmed, Nazia; Baldwin, David; Booton, Richard; Cochrane, Ann; Darby, Michael; Franks, Kevin; Hinde, Sebastian; Janes, Sam; Macleod, Una; Messenger, Mike; Moller, Henrik; Murray, Rachael; Neal, Richard; Quaife, Samantha; Sculpher, Mark; Tharmanathan, Puvanendran; Torgerson, David; CALLISTER, Matthew

VERSION 1 – REVIEW

REVIEWER	John Brodersen University of Copenhagen, Denmark
REVIEW RETURNED	20-Apr-2020

GENERAL COMMENTS	I have read the manuscript and my worries are that the suggestion of a new RCT comparing LDCT screening for lung cancer with usual care is not justified. A new RCT should try and answer those research questions that have not been addressed or have been raised in previous research. I have very recently published a rapid review about the degree of overdiagnosis in LDCT screening for lung cancer.[1] In our review we write the following in the paragraph for implication for research: Implications for research Lung cancer screening leads to reduction in mortality while it also leads to additional harm due to overdiagnosis and a high number of false positives. Decision makers face a difficult situation trying to balance between benefits and harms of lung cancer screening. Firstly, because the decision whether “benefit outweighs harms” is value-laden and it likely varies between individuals and across cultural settings. Secondly, because tools to assist decision makers with balancing the benefits and harms of screening are sparse.[2, 3] Moreover, the balance between benefits and harms of LDCT lung cancer screening should be viewed with its direct and indirect costs for people and for society compared to public health alternatives such as primary tobacco prevention. Researchers need to assess potential ways of minimizing harms of LDCT screening, especially the degree of overdiagnosis. The source of the variation in lung cancer overdiagnosis rates is currently unknown but should be the focus of future research to better understand what causes it and how to prevent it if lung screening programmes are
--

	implemented. Sources of possible variation that could be investigated include: variation in the population screened (age group screened, definition of heavy smokers, other risks specific to the screened population including e.g. asbestos, genomic variability etc.); differences in screening practice (screening intervals; numbers of screening rounds; differences in screening technology, differences in how abnormal findings are managed); and professional differences (radiological and pathological thresholds). A first potential strategy to reduce overdiagnosis is changing the eligibility criteria for LDCT screening. So far, participants were selected based on age and smoking history. Now there are risk prediction models that consider other variables and may be better at increasing the benefits and decreasing the harms of LDCT screening. However, high quality RCTs are needed to give valid answers to such research questions. The second strategy is changing the frequency of screening, the criteria for recall to further investigation, and thresholds to start or to stop screening. Modelling studies are exploring the consequences of changing screening parameters. Again, high quality evidence from RCTs is needed. Given the trade-offs between benefits and harms, researchers need to consider how decision aids may be incorporated into screening programmes to help healthy heavy smokers (current and former) make an informed choice about whether to participate in CT-screening for lung cancer. The present RCT described in the protocol submitted to BMJ Open will not be able to answer these questions because the RCT is not designed to do so. Moreover, the QoL is measured with generic measures which are inadequate according to best available evidence. Condition-specific measures should be used.[4] Kind regards John Brodersen  1. Brodersen et al. Overdiagnosis of lung cancer with low-dose computed tomography screening: meta-analysis of the randomised clinical trials. Breathe, 2020, 16(1): 200013. 2. Kinsinger et al. Implementation of Lung Cancer Screening in the Veterans Health Administration. JAMA Intern Med 2017, 177(3):399-406. 3. Caverly et al. Comparison of Observed Harms and Expected Mortality Benefit for Persons in the Veterans Health Affairs Lung Cancer Screening Demonstration Project. JAMA Internal Medicine 2018, 178(3):426-428. 4. Kauczor et al. ESR/ERS statement paper on lung cancer screening. Eur Radiol. 2020.
--	--

VERSION 1 – AUTHOR RESPONSE

We have responded to the reviewer’s comments below.

I have read the manuscript and my worries are that the suggestion of a new RCT comparing LDCT screening for lung cancer with usual care is not justified. A new RCT should try and answer those research questions that have not been addressed or have been raised in previous research.

We thank the reviewer for taking time to consider our protocol for publication. We suggest that the protocol does address new and relevant research questions and highlight two specific questions below.

1) Does lung cancer screening improve lung cancer outcomes at a population level?

Previous lung cancer screening studies have randomized recruited participants between screening and usual care. Previous communication with the Independent Chair of the UK National Screening Committee has indicated the perceived need for demonstration of benefit of screening at population

level - that is for the whole population invited to screening (including those who did not respond to invitation, or who were ineligible) compared to a non-invited usual care population. Reference has been made to a previous 'Zelen's design' study of Colorectal Screening published in 1996 (Hardcastle et al, Lancet 1996;348:1472-1477) in which improvements in outcomes from Colorectal Cancer were demonstrated across the whole population invited to a screening programme. YLST also uses a Zelen's design and aims to provide evidence on the efficacy of screening at a population level.

2) Do risk prediction models perform better in selecting a population for screening compared to criteria based on age and smoking history alone?

There is no current published prospective comparison of the performance of lung cancer risk prediction models in selecting a population for screening compared to criteria based on age and smoking alone (as used in the National Lung Screening Trial). The proposed study prospectively compares the PLCO and LLP models against the NLST criteria (extended to 80 years as per the US Preventive Services Task Force recommendation).

The second paragraph in the reviewer's comments is pasted from his recent review of Lung Cancer Screening. This paragraph covers many topics; we have highlighted several areas he raises where we suggest our protocol will provide relevant evidence.

Researchers need to assess potential ways of minimizing harms of LDCT screening, especially the degree of overdiagnosis. The source of the variation in lung cancer overdiagnosis rates is currently unknown but should be the focus of future research to better understand what causes it and how to prevent it if lung screening programmes are implemented. Sources of possible variation that could be investigated include: variation in the population screened (age group screened, definition of heavy smokers, other risks specific to the screened population including e.g. asbestos, genomic variability etc.); differences in screening practice (screening intervals; numbers of screening rounds; differences in screening technology, differences in how abnormal findings are managed); and professional differences (radiological and pathological thresholds).

We agree with the reviewer that overdiagnosis is a vitally important consideration for screening programmes. In order to estimate rates of overdiagnosis, studies must incorporate a non-screened control group against which to compare overall rates of cancer (both screen and non-screen detected). The inclusion of a usual care group in YLST will permit an estimate of overdiagnosis between intervention and control groups (indeed our study includes a non-invited control arm and thus will provide further information on this issue). We agree that differences in how abnormal findings are managed and radiological thresholds are of particular importance in overdiagnosis. Analysis of overdiagnosis in NLST has suggested that Bronchioalveolar Cell Carcinoma (which often presents as sub-solid nodules or SSNs) are a major cause of overdiagnosis in Lung Cancer Screening. The protocol includes an algorithm for managing SSNs in YLST that is more conservative than those currently used in the UK (the BTS Pulmonary Nodule Guidelines) and thus will hopefully reduce the rate of overdiagnosis accordingly.

A first potential strategy to reduce overdiagnosis is changing the eligibility criteria for LDCT screening. So far, participants were selected based on age and smoking history. Now there are risk prediction models that consider other variables and may be better at increasing the benefits and decreasing the harms of LDCT screening. However, high quality RCTs are needed to give valid answers to such research questions.

We agree that risk prediction models may allow better targeting of LDCT screening at an at-risk population, and that these models need prospective comparison in trials. As described above, comparison of two of the most studied risk-prediction tools (PLCO and LLP) against the criteria used in the National Lung Screening Trial is a primary endpoint for our study. The reviewer may be suggesting randomizing participants between these different tools (e.g. one population screened according to USPSTF criteria versus another population screened according to a PLCO threshold). This was an option we explored, but the required size of the study to power such comparison was prohibitively large. The option described in the protocol (whereby participants qualify for a screen based on fulfilling any of the three criteria, with subsequent comparison of the yield of each) allows a powered comparison between these criteria with much smaller numbers of recruits. Two other ongoing studies (the International Lung Screening Trial and the SUMMIT-Grail study) are using the same approach to prospectively compare PLCO with USPSTF.

The second strategy is changing the frequency of screening, the criteria for recall to further investigation, and thresholds to start or to stop screening. Modelling studies are exploring the consequences of changing screening parameters.

We agree that these are all important research questions, which are not addressed in YLST. However we suggest that a single trial would be unable to address all the questions listed in this paragraph.

The present RCT described in the protocol submitted to BMJ Open will not be able to answer these questions because the RCT is not designed to do so. Moreover, the QoL is measured with generic measures which are inadequate according to best available evidence. Condition-specific measures should be used.[4]

We agree with the reviewer that the submitted protocol does not answer all the questions raised in his recent review. We suggest though that it answers some of the questions he has identified (on overdiagnosis and risk prediction scores) together with other important research

questions (evidence of benefit at a population level) described above. The Quality of Life scores used (SF-12, EQ5D) are generic and allow comparison with other screening trials.

As previously discussed, we are almost two thirds of the way through recruitment to the study, and are therefore unable to alter the protocol at this stage in response to these peer-review comments. We would be delighted if BMJ Open was able to publish this unamended protocol, to appear alongside the linked study, the Yorkshire Enhanced Stop Smoking study. We would be grateful if you could clarify whether these reviewer comments preclude this.